# ALIGN-DEFORM-SUBTRACT: AN INTERVENTIONAL FRAMEWORK FOR EXPLAINING OBJECT DIFFERENCES

**Cian Eastwood**[*]
School of Informatics
University of Edinburgh
c.eastwood@ed.ac.uk

**Li Nanbo**[*]
School of Informatics
University of Edinburgh
nanbo.li@ed.ac.uk

**Christopher K. I. Williams**
School of Informatics
University of Edinburgh
ckiw@inf.ed.ac.uk

## ABSTRACT

Given two object images, how can we *explain* their differences in terms of the underlying object properties? To address this question, we propose *Align-Deform-Subtract* (ADS)—an interventional framework for explaining object differences. By leveraging semantic alignments in image-space as counterfactual interventions on the underlying object properties, ADS iteratively quantifies and removes differences in object properties. The result is a set of "disentangled" error measures which explain object differences in terms of the underlying properties. Experiments on real and synthetic data illustrate the efficacy of the framework.

## 1 INTRODUCTION

Given two object images, such as those depicted in Fig. 1a, can we explain *how* the objects in those images differ in terms of their properties? For example, can we say that the objects depicted in Fig. 1a have similar poses but differ in terms of shape and appearance? Interpretable explanations of *how* two object-images differ, in terms of their properties, has important applications in image retrieval (Wan et al., 2014), assessing goodness-of-fit (Belin & Rubin, 1995; Gelman et al., 2004), and learning "disentangled" generative models (Kulkarni et al., 2015; Eastwood & Williams, 2018).

One simple approach might be to just report a single-number summary of the object differences, such as the *mean squared error* (MSE). However, such summaries are clearly insufficient as they fail to tell us *how* or *why* the objects are different. Another approach would be to manually annotate (tens of) thousands of object-image pairs with their ground-truth property differences. However, since 1000 images have $P(1000, 2) = 999000$ possible pairings, this seems prohibitively expensive.

To address this problem, we propose *Align-Deform-Subtract* (ADS)—an interventional framework for explaining object differences. ADS takes a causal perspective on the above problem, asking *what must we do to object 1 (source) so that it looks like object 2 (target)*, or, equivalently, *how must we change or* intervene *upon the properties of the source object such that it looks like the target object?* To intervene without an explicit object model, ADS leverages the image-space transformations of semantic alignment networks (Rocco et al., 2018a) as counterfactual interventions on the underlying object properties. By iteratively intervening on the source-object properties, setting them to be those of the target, ADS both quantifies and removes object-property differences. The result is a set of disentangled error measures explaining *how* the objects differ in terms of their underlying properties.

---

[*]Equal contribution.

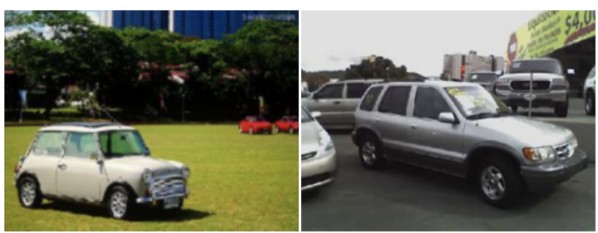

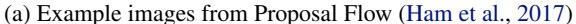

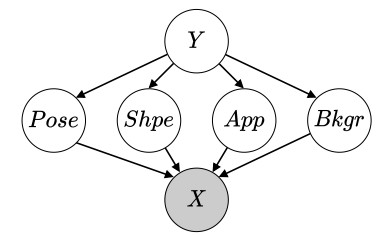

(a) Example images from Proposal Flow (Ham et al., 2017)  (b) Causal model

Figure 1: (a) *How* do these objects differ? A single-number summary like MSE fails to answer this question. In this work, we seek more fine-grained errors or *explanations* of object differences. (b) In particular, we seek to explain differences in terms of pose, shape and appearance. $Y$ denotes class, $X$ the observed image.

Figure 2: ADS overview. The properties of the source object are iteratively aligned with those of the target (left to right). *1) Align:* An aligning affine transformation $T_{\text{aff}}$ is used to intervene on the pose of the source object, setting it to that of the target. Importantly, $T_{\text{aff}}$ both removes and quantifies (via its magnitude) pose differences. *2) Deform:* An aligning TPS transformation $T_{\text{tps}}$ is then used to intervene on the shape of the source object, setting it to that of the target. Importantly, $T_{\text{tps}}$ both removes and quantifies (via its magnitude) shape differences. *3) Subtract:* Finally, the MSE between the aligned, deformed and source-masked images quantifies the appearance difference.

## 2 BACKGROUND: SEMANTIC ALIGNMENT NETWORKS

Estimating correspondences between images is an important problem in computer vision (Hartley & Zisserman, 2003). The classical approach has been to combine several different procedures like detecting and matching local features (e.g. SIFT, Lowe 2004), pruning incorrect matches using local geometric constraints (Schmid & Mohr, 1997), and estimating a global geometric transformation (Fischler & Bolles, 1981). However, recent works have achieved state-of-the-art results using *convolutional neural networks* (CNNs) as an end-to-end solution for semantic alignment (Rocco et al., 2018a;b). In particular, Rocco et al. (2018a) introduce a CNN architecture which takes as input two images—a *source* image $X_s$ and a *target* image $X_t$—and outputs the parameters of a geometric transformation which aligns them. To best align source and target images, Rocco et al. use the following two-stage procedure:

1. *Affine alignment:* A CNN-based alignment network $f_{\text{aff}}$ estimates the parameters $\theta_{\text{aff}}$ of the affine transformation $T_{\text{aff}}$ best-aligning the source and target images. More concretely, $\theta_{\text{aff}} = f_{\text{aff}}(X_s, X_t)$ and $T_{\text{aff}} = \arg\min_{\tau \in \mathcal{T}_{\text{affine}}} D_{\text{keypoints}}(\tau(X_s), X_t)$, with $D_{\text{keypoints}}$ being some measure of the "distance" between keypoints. As depicted in the second column of Fig. 2, $T_{\text{aff}}$ is then applied to $X_s$. Fig. 10 of Appendix D depicts keypoint "distances" before and after alignment.

2. *Thin-plate spline (TPS) alignment:* To refine the rough affine alignment, a second network $f_{\text{tps}}$ is used to estimate the parameters $\theta_{\text{tps}}$ of the TPS transformation $T_{\text{tps}}$ best-aligning the *affine-aligned* source and target images, i.e. $T_{\text{tps}} = \arg\min_{\tau \in \mathcal{T}_{\text{TPS}}} D_{\text{keypoints}}(\tau(T_{\text{aff}}(X_s)), X_t)$, and $\theta_{\text{tps}} = f_{\text{tps}}(T_{\text{aff}}(X_s), X_t)$. $T_{\text{tps}}$ is then applied to $T_{\text{aff}}(X_s)$, as depicted in the third column of Fig. 2.

Importantly, this method is fully trainable from synthetic transformations without the need for manual annotations, and generalizes to unseen images. See Rocco et al. (2018a) for more details.

## 3 INTERVENTIONAL FRAMEWORK

We now describe ADS—our interventional framework for explaining object differences. As depicted in Fig. 2, source-object properties are iteratively aligned with that of the target in a 3-step procedure, with each step both quantifying and removing a particular object-property difference. By viewing these image-space transformations as (counterfactual) interventions on the source-object properties, we can precisely describe our framework and its assumptions using the language of causal inference.

### 3.1 ALIGN

First, we leverage a pretrained affine alignment network $f_{\text{aff}}$ (see § 2) to obtain the parameters of the affine transformation $T_{\text{aff}}$ best-aligning our source and target images, i.e. $\theta_{\text{aff}} = f_{\text{aff}}(X_s, X_t)$. We then view $T_{\text{aff}}(X_s)$—depicted in the second column of Fig. 2—as a counterfactual intervention on the pose of the source object, setting it to be that of the target object. We denote this counterfactual— *"what if the source object had the pose of the target object?"*—as $X_s^{\mathcal{C};do(\text{Pose}:=\text{pose}_t)}$, with $\mathcal{C}$ a *structural causal model* (SCM, Pearl 2009) consistent with Fig. 1b and $X_s$ a slight abuse of notation to denote $X = x_s$ (i.e. conditioning on the factually-observed source object). Armed with this causal perspective on aligning transforms, we now describe the two key insights which allow us explain object-pose differences using $T_{\text{aff}}$:

1. $T_{\text{aff}}$ **quantifies the pose differences**. Since $T_{\text{aff}}(X_s) \approx X_s^{\mathcal{C};do(\text{Pose}:=\text{pose}_t)}$, we note that $||T_{\text{aff}}|| \propto \Delta_{\text{pose}}(X_s, X_t)$, where $|| \cdot ||$ is some measure of transformation magnitude and $\Delta_{\text{pose}}$ the ground-truth pose difference. Intuitively, $||T_{\text{aff}}||$ is the "effort" of transforming the source-object pose into that of the target. Furthermore, by decomposing $T_{\text{aff}}$ into scale, translation, and rotation components, we can get more fine-grained measures of object differences—namely *estimated* scaling $\hat{s}$, translation $\hat{t}$ and in-plane rotation $\hat{\theta}$. See Appendix A for details, Table 2 for a summary.

2. $T_{\text{aff}}$ **removes the pose differences**. Since $T_{\text{aff}}(X_s) \approx X_s^{\mathcal{C};do(\text{Pose}:=\text{pose}_t)}$, it follows that $\Delta_{\text{pose}}(T_{\text{aff}}(X_s), X_t) \approx \Delta_{\text{pose}}(X_s^{\mathcal{C};do(\text{Pose}:=\text{pose}_t)}, X_t) = 0$. This removal or explaining-away of pose errors is critical for subsequent steps which assume that no pose differences remain.

## 3.2 DEFORM

Next, we leverage a pretrained TPS alignment network $f_{\text{tps}}$ (see § 2) to obtain the parameters of the TPS transformation $T_{\text{tps}}$ best-aligning our (affine-aligned) source and target images, i.e. $\theta_{\text{tps}} = f_{\text{tps}}(T_{\text{aff}}(X_s), X_t)$. We then view $T_{\text{tps}}(T_{\text{aff}}(X_s))$—depicted in the third column of Fig. 2—as a further (counterfactual) intervention on the source object, setting its shape to be that of the target object. We denote this counterfactual—*"what if the source object also had the shape of the target object?"*—as $X_s^{\mathcal{C};do(\text{Pose}:=\text{pose}_t,\text{Shape}:=\text{shape}_t)}$. As before, this causal perspective on aligning transforms makes clear two key insights which allow us to explain shape differences with $T_{\text{tps}}$:

1. $T_{\text{tps}}$ **quantifies the shape differences**. Since $T_{\text{tps}}(T_{\text{aff}}(X_s)) \approx X_s^{\mathcal{C};do(\text{Pose}:=\text{pose}_t,\text{Shape}:=\text{shape}_t)}$, we note that $||T_{\text{tps}}|| \propto \Delta_{\text{shape}}(X_s, X_t)$, where $||T_{\text{tps}}||$ is seen as the "effort" of transforming the source-object shape into that of the target (see Appendix A for details). This gives us an estimate of the shape or deformation difference $\hat{d}$. Note that, if we had not removed pose differences in the previous step, $||T_{\text{tps}}||$ would be an "entangled" measure of both pose *and* shape differences.

2. $T_{\text{tps}}$ **removes the shape differences**. Since $T_{\text{tps}}(T_{\text{aff}}(X_s)) \approx X_s^{\mathcal{C};do(\text{Pose}:=\text{pose}_t,\text{Shape}:=\text{shape}_t)}$, it follows that $\Delta_{\text{shape}}(T_{\text{tps}}(T_{\text{aff}}(X_s)), X_t) \approx \Delta_{\text{shape}}(X_s^{\mathcal{C};do(\text{Pose}:=\text{pose}_t,\text{Shape}:=\text{shape}_t)}, X_t) = 0$. This removal or explaining-away of shape errors is critical for the final step which assumes no pose or shape differences remain.

## 3.3 SUBTRACT

Finally, after removing pose and shape differences using $T_{\text{aff}}$ and $T_{\text{tps}}$ respectively, we seek to quantify appearance differences. Assuming the causal model of Fig. 1b, only appearance and background differences remain. To remove the latter, we make use of an estimated source-object mask $\tilde{M}_s$, obtained from a pretrained instance segmentation network (He et al., 2017) or otherwise.[1] Writing this background removal more formally, we have $\Delta_{\text{backgr.}}(\tilde{M}_s \odot T_{\text{tps}}(T_{\text{aff}}(X_s)), \tilde{M}_s \odot X_t)) \approx \Delta_{\text{backgr.}}(X_s^{\mathcal{C};do(\text{Pose}:=\text{pose}_t,\text{Shape}:=\text{shape}_t,\text{Backgr.}:=0)}, X_t^{\mathcal{C};do(\text{Backgr.}:=0)}) = 0$, where $\odot$ denotes element-wise multiplication. We can then get an estimated appearance difference $\hat{a}$ using a masked MSE between the affine- and TPS-aligned source image $T_{\text{tps}}(T_{\text{aff}}(X_s))$ and the target image $X_t$, i.e. $\hat{a} = \text{MSE}(\tilde{M}_s \odot T_{\text{tps}}(T_{\text{aff}}(X_s)), \tilde{M}_s \odot X_t)$. The penultimate column of Fig. 2 provides a visualisation of $\hat{a}$ *before* averaging over unmasked foreground pixels, using a square-error heatmap. Note that the standard MSE—$\text{MSE}(X_s, X_t)$—*entangles* errors arising from differences in pose, shape, appearance *and* background, while our measures *disentangle* these errors through iterative interventions on the source object. This crucial difference is illustrated in Fig. 7 of Appendix C.1.

## 4 EXPERIMENTS

### 4.1 REAL DATA: QUALITATIVE EVALUATION

We first evaluate our measures on real data from the *Proposal Flow* (PF) dataset (Ham et al., 2017). As the *ground-truth* (GT) property differences are not available, this evaluation is *qualitative*, i.e. one must decide, by visual inspection, how well our measures explain the object-property differences.

---

[1] If the source image is the rendering of an explicit object model, we would get the mask $\tilde{M}_s$ "for free".

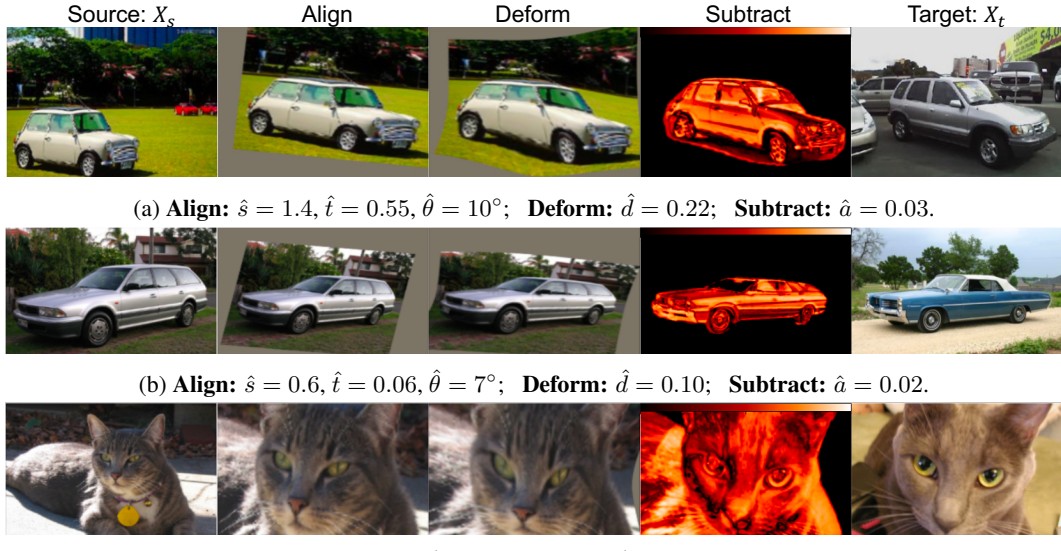

| Source: $X_s$ | Align | Deform | Subtract | Target: $X_t$ |

(a) **Align:** $\hat{s} = 1.4$, $\hat{t} = 0.55$, $\hat{\theta} = 10°$; **Deform:** $\hat{d} = 0.22$; **Subtract:** $\hat{a} = 0.03$.

(b) **Align:** $\hat{s} = 0.6$, $\hat{t} = 0.06$, $\hat{\theta} = 7°$; **Deform:** $\hat{d} = 0.10$; **Subtract:** $\hat{a} = 0.02$.

(c) **Align:** $\hat{s} = 4.5$, $\hat{t} = 0.27$, $\hat{\theta} = 6°$; **Deform:** $\hat{d} = 0.16$; **Subtract:** $\hat{a} = 0.08$.

Figure 3: Our measures on the PF dataset. Table 2 of Appendix A summarises our measures and their units.

**Results.** By comparing the relative magnitude of our measures across the 3 examples in Fig. 3, we see that our measures accurately capture the object-property differences. For example, letting $\hat{s}^{(a)}$ denote our estimated scaling factor $\hat{s}$ in Fig. 3a, we see that $\hat{s}^{(c)} > \hat{s}^{(a)} > \hat{s}^{(b)}$, which is visually consistent with source-object scalings. Likewise, $\hat{t}^{(a)} > \hat{t}^{(c)} > \hat{t}^{(b)}$ is consistent with the source-object translations, as $|\hat{\theta}^{(a)}| > |\hat{\theta}^{(b)}| > |\hat{\theta}^{(c)}|$ is with the rotation magnitudes.

## 4.2 SYNTHETIC DATA: QUANTITATIVE EVALUATION

We next evaluate our measures on synthetic data where we have the GT object differences. To do so, we use the teapot model of Moreno et al. (2016) to generate 2000 images of teapots, randomly sampling pose, shape, appearance and background parameters. We then randomly pair these images, resulting in 1000 source-target pairs, and calculate the GT differences in pose, shape and appearance. Appendix B details the generation process and subsequent calculation of GT "distances" between teapot properties, while also depicting some example images (see Fig. 5). Next, we pass all source-target pairs $\{(X_s^{(i)}, X_t^{(i)})\}_{i=1}^{1000}$ through affine and TPS alignment networks—pretrained on PF—to get $\theta_{\text{aff}}^{(i)} = f_{\text{aff}}(X_s^{(i)}, X_t^{(i)})$ and $\theta_{\text{tps}}^{(i)} = f_{\text{aff}}(T_{\text{aff}}^{(i)}(X_s^{(i)}), X_t^{(i)})$ (see Fig. 8 of Appendix C.2 for example transformations). Finally, we use $\theta_{\text{aff}}^{(i)}$ and $\theta_{\text{tps}}^{(i)}$ to calculate our pose ($\hat{s}_x, \hat{s}_y, \hat{t}_x, \hat{t}_y, \hat{\theta}$), shape ($\hat{d}$), and appearance ($\hat{a}$) measures for each source-target pair (see Appendix A).

**Results.** Each panel in Fig. 4 comprises two plots. The blue dots show our predicted object-property difference (y-axis) against the GT object-property difference (x-axis). The yellow dots show the MSE for a particular object-property difference—as MSE is also affected by differences in all other object properties, it is poorly correlated with the particular object-property difference against which it is plotted. For example, the bottom-left panel plots the GT difference in $x$-position, $t_x$, against our prediction $\hat{t}_x$ (blue dots) and the MSE (yellow dots). Evidently, the GT difference has a much stronger correlation with our prediction than it does with the MSE—as made clear by the correlation coefficients in Table 1. Importantly, the GT appearance difference $a$ has a much stronger correlation with our prediction $\hat{a}$ than it does with the MSE. This improvement is explained by Fig. 9—by first intervening on the pose and shape of the source object such that they match those of the target, our measure gains invariance to pose and shape differences. Overall, most of our measures show a strong correlation with the corresponding GT object-property difference, indicating that they accurately capture the mismatch for a particular object property while remaining relatively invariant to mismatches in other object properties. This confirms that our ADS framework can indeed disentangle errors arising from different object-property mismatches, and, as a result, provide a good explanation of *how* the source and target objects differ. One notable exception is shape ($d$). We attribute this to the TPS network of Rocco et al. (2018a), which often performs quite poorly. We hope that this will be resolved by future improvements in semantic alignment networks.

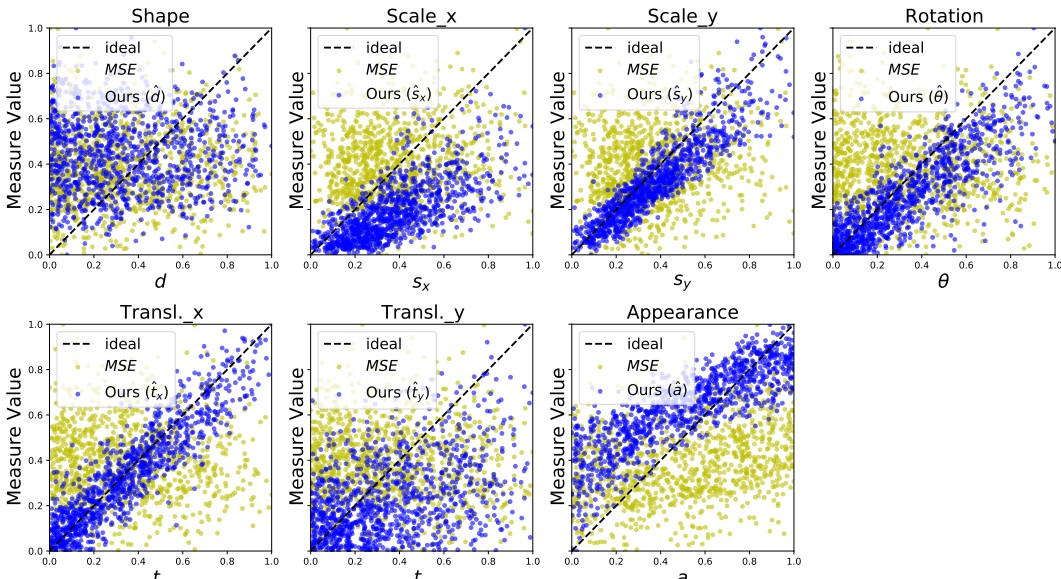

Figure 4: Ground-truth (GT) object-property differences vs. our measures on the Teapot dataset. Note that we standardise both the GT values and our measures to lie in $[0, 1]$ and use a log-log plot for appearance.

Table 1: Pearson correlation coeff. between GT differences and: (i) MSE; (ii) our corresponding measure.

|      | $d$    | $s_x$  | $s_y$  | $\theta$ | $t_x$  | $t_y$  | $a$    |
|------|--------|--------|--------|----------|--------|--------|--------|
| MSE  | **0.03** | 0.14   | 0.14   | 0.01     | 0.01   | 0.04   | 0.33   |
| Ours | 0.01   | **0.71** | **0.90** | **0.85** | **0.91** | **0.41** | **0.84** |

## 5 DISCUSSION

**Related work.** Belongie et al. (2002) quantified object shape differences using the magnitude of the aligning transform along with the sum of the matching errors between correspondence points. In self-supervised learning, Von Kügelgen et al. (2021) and Mitrovic et al. (2021) recently viewed image transformations or augmentations as counterfactual interventions on underlying properties.

**Use cases.** Potential use cases for interpretable, fine-grained explanations of object differences include: (i) *Image retrieval:* our measures could permit the retrieval of object images that are most similar in pose, shape *or* appearance; (ii) *Assessing goodness-of-fit:* If the source image is the output or rendering of an object model, then our measures elucidate *how* and *why* the model is wrong. While "all models are wrong" (Box, 1976) in practice, these measures at least allow us to determine if the model is useful (Belin & Rubin, 1995, pg. 165), e.g. manufacturing applications may care exclusively about object shape; and (iii) *Learning disentangled generative models*: Disentangled errors may aid the learning of disentangled generative models (Kulkarni et al., 2015) in which pose, shape and appearance are represented separately.

**Limitations.** We make several simplifying assumptions that may be seen as limitations: (i) *Simple causal model:* In reality, Fig. 1b may not always be correct, e.g. there could be other object properties, or multiple objects in one image; (ii) *Alignment networks work well:* Our framework and its constituent measures rely on good (rigid and non-rigid) alignment. While this may not be true for all datasets, such networks will only improve in the future (semantic alignment is a popular topic in the computer vision community); (iii) *Pose primarily differs in the image plane:* To handle large 3D pose errors about any axis, future work may look to incorporate explicit object models.

**Conclusion.** We have presented ADS—an interventional framework for explaining object differences. ADS leverages the image-space transformations of semantic alignment networks to *align* and *deform* the source-object image such that it matches the target, and casts these image-space alignments as counterfactual interventions on the underlying object properties. This allows ADS to disentangle errors arising from different object-property mismatches, ultimately providing fine-grained explanations of object differences in terms of their underlying properties. We believe that ADS constitutes an interesting and novel framework, even if some components need improvement before it can be deployed in the real-world.

ACKNOWLEDGEMENTS

We thank Robert Fisher, Julius von Kügelgen and Andrew Zisserman for helpful discussions and comments. We also thank Pol Moreno for help with the generation of the teapot dataset.

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

## A  CALCULATION OF OUR MEASURES

We now detail the calculation of our measures. Similar to Belongie et al. (2002), we leverage the magnitude of the aligning transform to quantify object-property dissimilarity, viewing this magnitude as the "effort" of transforming the source-object property into that of the target.

**Affine transform.**  For the affine transform, we have:

$$X_{\text{aff}}^{\text{grid}} = \begin{bmatrix} r_{11} & r_{12} \\ r_{12} & r_{22} \end{bmatrix} X_s^{\text{grid}} + \begin{bmatrix} \hat{t}_x \\ \hat{t}_y \end{bmatrix}, \tag{1}$$

where $X_s^{\text{grid}}, X_{\text{aff}}^{\text{grid}} \in \mathbb{R}^{2 \times N}$ ($N$ is the number of pixels) denote the grid coordinates of the source and transformed-source images respectively. Following Rocco et al. (2018a), the six affine parameters $\theta_{\text{aff}} = [r_{11}, r_{12}, r_{21}, r_{22}, \hat{t}_x, \hat{t}_y]$ are the output of a pretrained affine-alignment network $f_{\text{aff}}$ (see § 2). To get more fine-grained measures, we further decompose this affine transform as:

$$\begin{bmatrix} r_{11} & r_{12} \\ r_{12} & r_{22} \end{bmatrix} = \begin{bmatrix} \cos(\hat{\theta}) & -\sin(\hat{\theta}) \\ \sin(\hat{\theta}) & \cos(\hat{\theta}) \end{bmatrix} \begin{bmatrix} 1 & b \\ 0 & 1 \end{bmatrix} \begin{bmatrix} \hat{s}_x & 0 \\ 0 & \hat{s}_y \end{bmatrix}, \tag{2}$$

where $\hat{\theta}$ is a measure of the in-plane rotation difference between the source and target objects, $(\hat{s}_x, \hat{s}_y)$ are measures of the scaling *factors*, $(\hat{t}_x, \hat{t}_y)$ are measures of the translation differences along the $x$ (horizontal) and $y$ (vertical) axis respectively, and $b$ is a measure of the shear differences. To ease interpretation, we often a report single measure for the *scaling factor* $\hat{s} = \hat{s}_x \hat{s}_y$, which represents the factor by which the source-object area is scaled. We also report a single translation difference $\hat{t} = \sqrt{\hat{t}_x^2 + \hat{t}_y^2}$.

**TPS deformation.**  For the TPS deformation, we again follow Rocco et al. (2018a), with the pretrained TPS-alignment network outputting the parameters of a TPS deformation on a $3 \times 3$ grid (a set of 9 points). That is, $\theta_{\text{tps}} \in \mathbb{R}^{2 \times 9}$ is a matrix containing the 2D coordinates of each of the 9 grid points. We denote the TPS-transformed image grid as $X_{\text{tps}}^{\text{grid}} \in \mathbb{R}^{2 \times N}$ ($N$ is the number of pixels) and the magnitude of this transformation or *deformation* $\hat{d}$ (which we use as our shape measure). Specifically, $\hat{d}$ is the average distance moved by $X_{\text{tps}}^{\text{grid}}$ after normalising with the diagonal image-grid size or "distance" ($2\sqrt{2}$):

$$\hat{d} = \frac{1}{2\sqrt{2}\,N} \cdot ||(X_{\text{tps}}^{\text{grid}} - X_{\text{aff}}^{\text{grid}})||_{2,1}, \tag{3}$$

where $||\cdot||_{2,1}$ represents the $L_{2,1}$ norm of a matrix.

**Appearance.**  Finally, we measure the appearance difference $\hat{a}$ using a source-masked MSE, i.e. $\hat{a} = \text{MSE}(\tilde{M}_s \odot T_{\text{aff}}(X_s), \tilde{M}_s \odot X_t)$, where $\tilde{M}_s$ denotes the source-object mask. Note that $\hat{a} \in [0, 1]$ since the RGB pixel values are in $[0, 1]^3$.

Table 2: Summary of our measures.

| Measure | Property | Range | Descript. (unit) |
|---|---|---|---|
| $\hat{s}$ | Scale | $[0, \infty)$ | Object-area scaling factor (unitless) |
| $\hat{t}$ | Translation | $[0, 1]$ | Distance moved (proportion of pixels) |
| $\hat{\theta}$ | Rotation | $[-180, 180]$ | In-plane rotation angle (degrees) |
| $\hat{d}$ | Shape | $[0, 1]$ | Avg. distance moved (proportion of pixels) |
| $\hat{a}$ | Appearance | $[0, 1]$ | MSE with pixels in $[0, 1]$ |

# B  TEAPOT DATASET

**Overview**    The teapot dataset contains 1000 data samples, where each sample consists of a source image (with its associated mask) and a target image—see Fig. 5 below.

Source: image + mask                                    Target: image

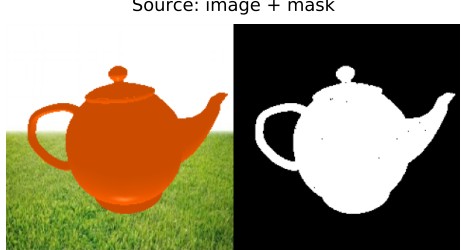 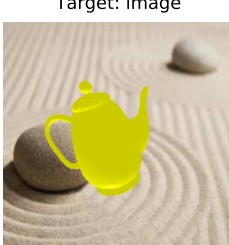

Figure 5: A teapot data sample.

**Generation.**    The generation of each teapot is controlled by a unique 7D vector, with each dimension describing a different property. Note that subscripts $x$ and $y$ used to indicate an item measured along the $x$ (horizontal) and $y$ (vertical) axis respectively. Thus, we store the 2000 7D vectors as object descriptors for the 2000 simulated teapots (1000 source + 1000 target).

**Simulated shape deformations.**    To simulate shape deformations while retaining access to a GT shape difference, we fit a PCA model to a set of different teapot mesh models, each consisting of 8163 vertices. We then use the top 10 principal components to describe a teapot mesh, which allows us to deform the shape by manipulating this 10D latent code. To simplify our experiments, we select a single dimension to manipulate while keeping fixed the other 9. As a result, the shape deformation of a teapot can be described using a single scalar $d \in \mathbb{R}$. We thus generate 2000 different teapots by randomly sampling 2000 latent values $d \sim \mathcal{U}(-4., 4.)$.

**Simulated geometric variations.**    As we set up our problem in the image domain where 3D spatial information is generally incomplete, we describe an object's geometric properties, specifically pose and scale, as 2D properties in the image plane. To simulate geometric variations (i.e. poses and scales) while retaining access to a GT difference, we describe the pose of a teapot using translations along the $x$ and $y$ axes, $t_x \in \mathbb{R}$ and $t_y \in \mathbb{R}$ respectively, and an in-plane rotation angle $\theta \in \mathbb{R}$ (clockwise). Similarly, we describe the scale of a teapot along the $x$ and $y$ axes as $s_x \in \mathbb{R}$ and $s_y \in \mathbb{R}$. During simulation, we noticed that the ratio $\frac{s_y}{s_x}$ can implicitly explain-away some of the shape variance. As a result, we fix $s_y = s_x$ when simulating the data. This allows us to interpret changes in scale as the effect of a change in distance between the object and camera. In sum, we randomly simulate 2000 pose parameters by sampling: $\theta \sim \mathcal{U}(-18°, 18°)$, $t_x \sim \mathcal{U}(-0.4, 0.4)$, $t_y \sim \mathcal{U}(-0.4, 0.4)$, and 2000 scale parameters by sampling: $s_x \sim \mathcal{U}(0.7, 1.3)$.

**Simulated appearance/texture changes.**    To simulate appearance changes while retaining access to a GT value, we interpolate RGB values between two randomly-selected texture maps using an interpolation ratio $a \sim \mathcal{U}(0, 1)$. For example, given two texture maps $\mathbf{A}_0$ and $\mathbf{A}_1$, a simulated texture is $a\mathbf{A}_1 + (1 - a)\mathbf{A}_0$. For simplicity, we only use texture maps with solid colours.

**Ground-truth "distance" measures.**    We compute the GT property differences for a source-target image-pair using the saved 7D object descriptors. Below we detail the calculation of these differences, using the subscripts "$src$" and "$tar$" to indicate properties of the source and target teapots:

- *Shape:* $d = |d^{tar} - d^{src}|$

- *Scale (2D):* $s_x = s_x^{tar}/s_x^{src}$ and $s_y = s_y^{tar}/s_y^{src}$

- *Rotation:* $\theta = \theta^{tar} - \theta^{src}$ (rotations defined within the image plane)

- *Translation (2D):* $t_x = t_x^{tar} - t_x^{src}$ and $t_y = t_y^{tar} - t_y^{src}$

- *Texture:* $a = |a^{tar} - a^{src}|$

## C  EXAMPLES

### C.1  PROPOSAL FLOW (PF)

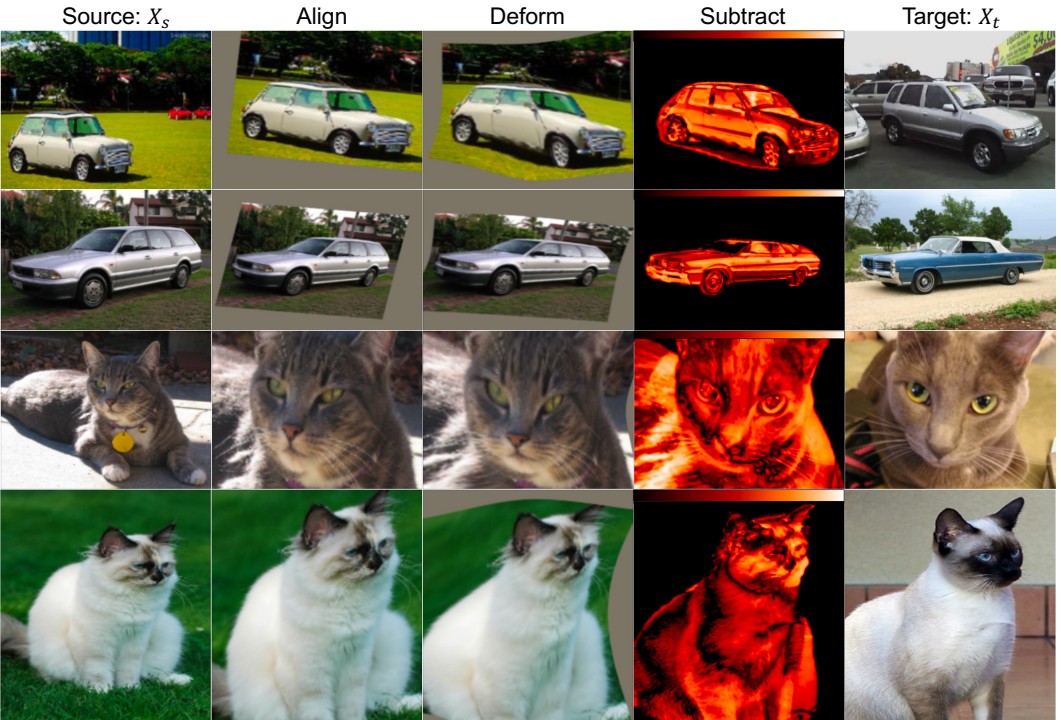

Figure 6: Examples from the PF dataset showing the 3 steps of our interventional framework: align, deform, and subtract.

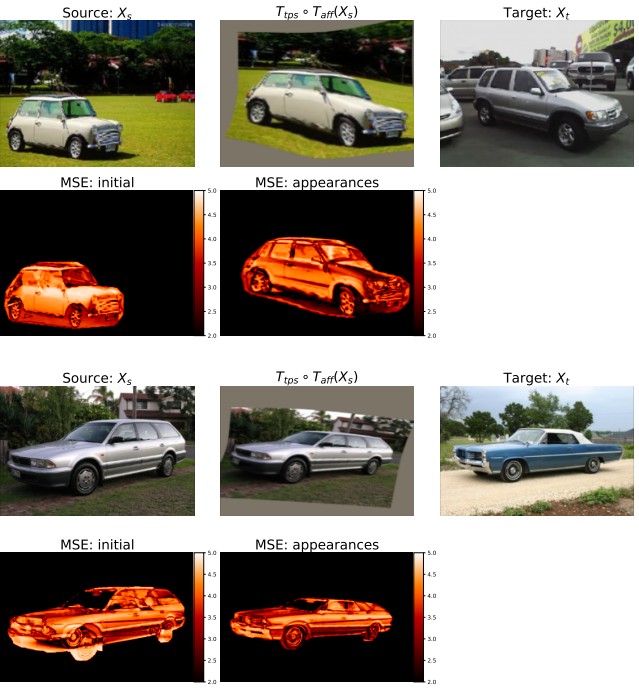

Figure 7: Examples from the PF dataset illustrating how: (i) the MSE initially captures errors arising from pose, shape *and* appearance differences (MSE: initial); and (ii) after alignment and deformation, the MSE only captures appearance errors (MSE: appearances).

## C.2 TEAPOTS

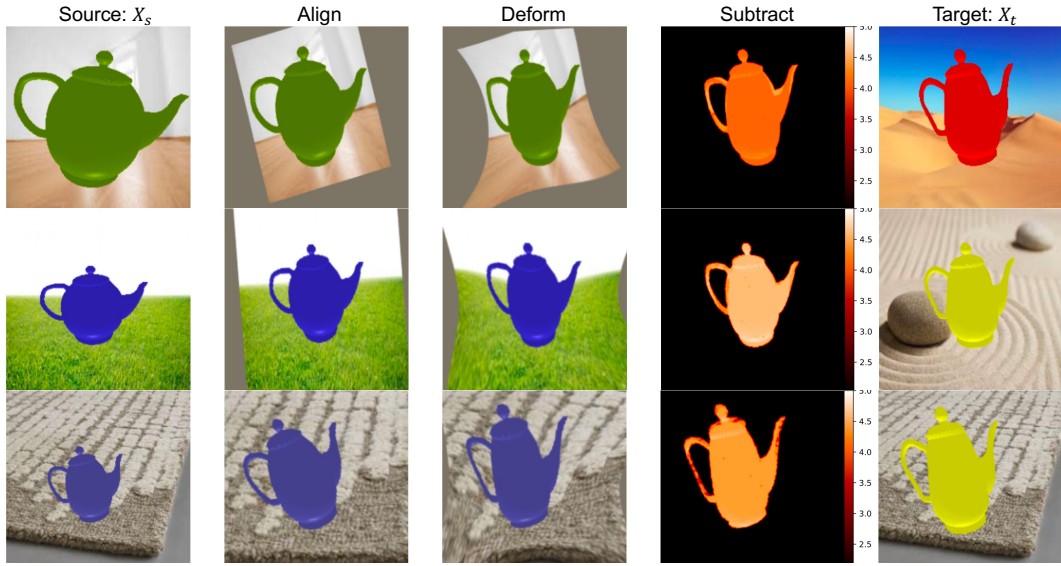

Figure 8: Examples from the Teapot dataset showing the 3 steps of our interventional framework: align, deform, and subtract.

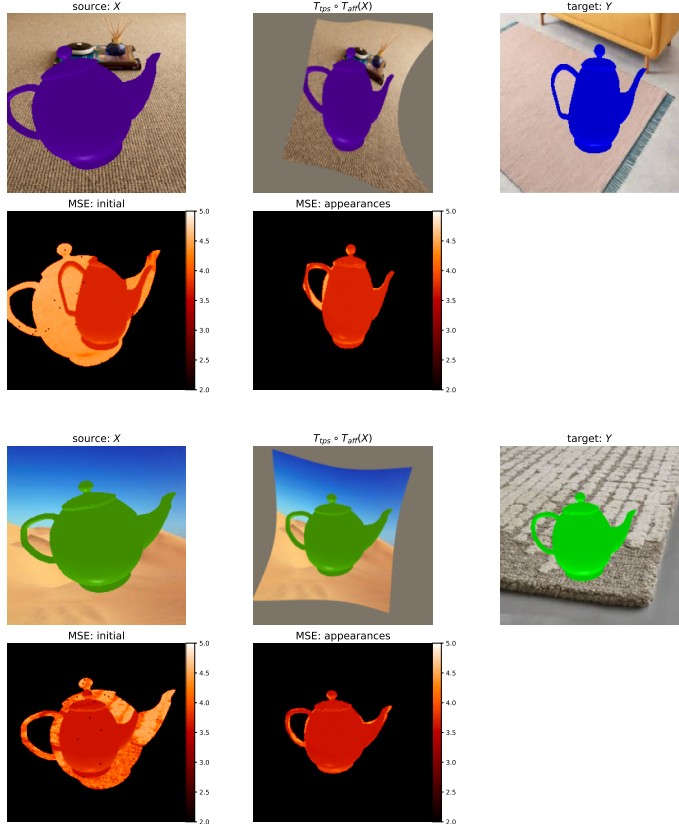

Figure 9: Teapot examples illustrating how: (i) the MSE initially captures errors arising from pose, shape *and* appearance differences (MSE: initial); and (ii) after alignment and deformation, the MSE only captures appearance errors (MSE: appearances).

## D   SEMANTIC ALIGNMENT NETWORKS

| Source: $X_s$ | $T_{aff}(X_s)$ | $T_{tps} \circ T_{aff}(X_s)$ | Target: $X_t$ |
| --- | --- | --- | --- |

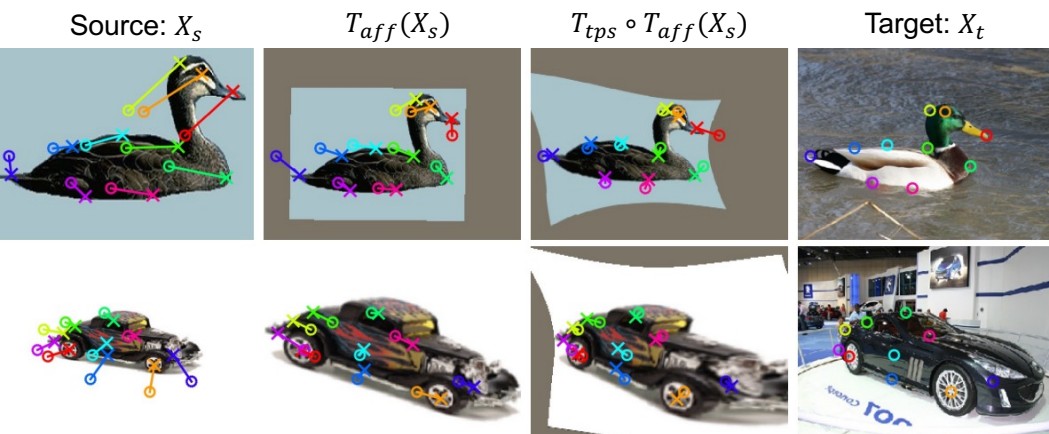

Figure 10: Figure and caption adopted from (Rocco et al., 2018a, Fig. 9). Each row shows one test example from the PF dataset (Ham et al., 2017). GT matching keypoints, only used for alignment evaluation, are depicted as crosses and circles for source $X_s$ and target $X_t$ images, respectively. Keypoints of same colour are supposed to match each other after $X_s$ is aligned to $X_t$. To illustrate the matching error, we also overlay keypoints of $X_t$ onto different alignments of $X_s$ so that lines that connect matching keypoints indicate the keypoint position error vector. The method manages to roughly align the images with an affine transformation $T_{\text{aff}}$ (column 2), and then perform finer alignment using $T_{\text{tps}}$ (column 3).

