# OpenReview forum: "Align-Deform-Subtract: An interventional framework for explaining object differences"
_ICLR.cc/2022/Workshop/OSC — ICLR2022 OSC  Poster_

### Official Review · Reviewer_yyzN · 2022-03-10
**A nice though somewhat simplistic framework for explaining object differences.**

**Rating:** 2
**Confidence:** 2

**Review:**

This paper proposes to use an alignment network output (from Rocco 2018) and MSE to quantify changes between two different images such that they "explain" the difference between the images via a "pose" change, a "shape" change and an "appearance" change. By using the outputs of the alignment network the authors propose a way to quantify pose changes (by magnitude of affine transform between images), shape changes (by magnitude of TPS deformation between images) and appearance changes (by a pixel wise L2 error).

My main issue, and the authors themselves admit it, is that this is a very simplistic way of explaining differences between images - it completely ignores the 3D aspect of natural images, and, even more crucially, the appearance model used is really too simple - other pre-trained features (maybe a self-supervised representation trained on ImageNet or something along these lines) could have been much more informative.

I like the core idea in this work and presenting at the workshop to get more feedback is probably a good idea, but I would encourage the authors to enrich and make the model a bit more sophisticated. The technology exists and would benefit this nice core idea.

---

### Official Review · Reviewer_Zeoy · 2022-03-15
**Accept**

**Rating:** 2
**Confidence:** 2

**Review:**

This paper is relevant for the workshop.
The authors explore which transforms need to be applied to a source object to match a target object.
The "Interventional Framework" section is described well.
It would be nice to see the qualitative results on the synthetic data.
In Figure 4: It is not clear how the MSE is computed? Also, to improve the paper for future submissions it would be worth comparing to some of the methods mentioned in the related work.

Why is the model better at scale y than scale x? Is this to do with bias in the data? Similar for translation x and translation y?
How do you disentangle depth from scale? Do you assume a fixed depth? This would be worth discussing.

---

### Decision · Program_Chairs · 2022-03-21

**Decision:**

Accept (Poster)

**Comment:**

The reviewers agree the paper should be accepted at the workshop. Congratulations!

The authors are encouraged to take the feedback by the reviewers into account when preparing the camera-ready version of the paper.